# Association between Urinary Cadmium-to-Zinc Intake Ratio and Adult Mortality in a Follow-Up Study of NHANES 1988–1994 and 1999–2004

**DOI:** 10.3390/nu12010056

**Published:** 2019-12-24

**Authors:** Kijoon Kim, Melissa M. Melough, Junichi R. Sakaki, Kyungho Ha, Dalia Marmash, Hwayoung Noh, Ock K. Chun

**Affiliations:** 1Department of Nutritional Sciences, University of Connecticut, Storrs, CT 06269, USA; drkijoon@gmail.com (K.K.); melissa.melough@uconn.edu (M.M.M.); junichi.sakaki@uconn.edu (J.R.S.); kyungho.ha@uconn.edu (K.H.); dalia.marmash@uconn.edu (D.M.); 2Department of Food and Nutrition, Sookmyung Women’s University, Seoul 04310, Korea; 3Nutritional Methodology and Biostatistics Group, Section of Nutrition and Metabolism, International Agency for Research on Cancer(IARC-WHO), 69372 Lyon, France; nohh@fellows.iarc.fr

**Keywords:** zinc intake, urinary cadmium, mortality, NHANES, urinary Cd/Zn intake ratio

## Abstract

Cadmium (Cd) is a toxic heavy metal associated with increased mortality, but the effect of zinc (Zn) intake on the association between Cd and mortality is unknown. The objective of this study was to examine the association of urinary Cd to Zn intake ratio (Cd/Zn ratio) and mortality risk. In total, 15642 US adults in NHANES 1988–1994 and 1999–2004 were followed until 2011 (15-year mean follow-up). Of the 5367 total deaths, 1194 were attributed to cancer and 1677 were attributed to CVD. After adjustment for potential confounders, positive associations were observed between urinary Cd and all-cause mortality (HR for highest vs. lowest quartile: 1.38; 95% CI: 1.14–1.68) and cancer mortality (HR: 1.54; CI: 1.05–2.27). Urinary Cd was positively associated with cancer mortality among the lowest Zn consumers, and the association diminished among the highest Zn consumers. Positive relationships were observed between the Cd/Zn ratio and all-cause mortality (HR: 1.54; CI: 1.23–1.93), cancer mortality (HR: 1.65; CI: 1.11–2.47) and CVD mortality (HR: 1.49; CI: 1.18–1.88). In conclusion, these findings indicate that Zn intake may modify the association between Cd and mortality. Furthermore, the Cd/Zn ratio, which was positively associated with mortality from all causes, cancer, and CVD, may be an important predictor of mortality.

## 1. Introduction

During certain industrial processes, cadmium (Cd), a highly toxic and carcinogenic metal, is produced as a byproduct. Cd can then leach into soil and water which are used in agricultural processes, causing Cd to appear in the food supply [1]. The food supply is the primary source of Cd exposure in non-smokers, and tobacco is an additional prime source of Cd in smokers [2]. Due to its long half-life, Cd may be accumulated in multiple tissue types in the human body, thereby leading to many diseases such as kidney dysfunction [3], reproductive dysfunction [4], cardiovascular disease [5], diabetes [6], osteoporosis [7], cancer [8], and increased mortality [9].

Higher Cd exposure has been linked to increased cardiovascular disease (CVD) risk in a Swedish cohort study [10] and linked to heart failure and mortality in elderly Australian women [11]. Cd exposure has also been associated with mortality from total, lung and pancreatic cancers in the Strong Heart Study, which included 4545 American Indian participants [12]. Several studies have reported that higher Cd exposure was associated with increased mortality from all causes, cancer, and CVD in the general populations of Japan [13], Belgium [14], and the US [15].

Interestingly, the mechanism underlying Cd’s effect on these diseases relates to its interaction with zinc (Zn). Cd competes with Zn in cellular uptake and metallothionein and due to its higher affinity for metallothionein, Cd displaces Zn, resulting in increased synthesis of metallothionein and Cd absorption [16]. Increased saturation of Cd at binding sites within the kidney can lead to microscopic tubular proteinuria (known to occur at urinary Cd concentrations of 2 µg/g creatinine) and renal dysfunction (at urinary Cd concentrations of 10 µg/g creatinine) [1,17]. Due to the competition between Cd and Zn for several key binding sites, dietary Zn may interfere in the absorption, accumulation, and bioavailability of Cd in the body. This has been demonstrated in several animal studies where Zn supplementation protected against Cd toxicity in the spleen, liver, and kidney [18,19,20]. Epidemiologic evidence indicates that Zn intake is linked to lower Cd burden in US adults, presumably due its competitive uptake and metallothionein binding with Cd [21]. Among US men with low Zn consumption and high concentrations of Cd excreted in urine, total prostate-specific antigen, a biomarker indicative of prostate cancer, was elevated, suggesting a relationship between dietary Zn, urinary Cd and cancer [22]. Cancer mortality risk appears to be higher among those with inadequate Zn intake [23].

These studies suggest that dietary Zn may protect against Cd toxicity in humans, indicating that dietary Zn should be considered in investigating the association between Cd exposure and health outcomes, including mortality. There are few studies exploring the relationship between the Cd/Zn ratio and mortality from all causes, cancer, and CVD. Therefore, the objective of this study was to investigate the association of the Cd/Zn ratio with mortality from all causes, cancer, and CVD among US adults.

## 2. Materials and Methods

### 2.1. Study Population

Data from 15,641 participants in NHANES 1988–1994 [24] and 1999–2004 [25,26,27] were used. Exclusion criteria consisted of pregnant or breastfeeding women (*n* = 458), those with unreliable or incomplete dietary recalls (*n* = 1128) and those with missing data on urinary creatinine and urinary Cd (*n* = 7957). Additionally, because cancer and CVD incidence and mortality rates are very low for young people [28,29], we excluded subjects under 30 years old. NHANES study protocols were approved by the National Center for Health Statistics research ethics review board. (Institutional Review Board approval and documented consent from participants for NHANES 1988–1994 and Protocol #98-12 for NHANES 1999–2004).

### 2.2. Estimation of Dietary Zn Intake

Dietary Zn intake was estimated from one 24-h dietary recall in NHANES 1988–1994 and 1999–2004.

### 2.3. Biomarker Data

Participants’ spot urine samples were collected in the mobile examination center. Urinary Cd was measured as described in the NHANES Laboratory Procedures Manual [30,31]. While elevated blood Cd confirms recent acute exposure, it does not correlate well with total body burden. Therefore, urinary Cd, which reflects integrated exposure and total body burden [32], was used in this analysis. Urinary Cd concentrations were determined using inductively coupled plasma-mass spectrometry (ICP-MS) and adjusted by creatinine.

### 2.4. Mortality Data Linked to the NHANES

We combined NHANES data from the survey years 1988–1994 and 1999–2004 with their linked mortality datasets. During a mean of follow-up of 15 years (through 31 December 2011), 5367 deaths occurred [33]. NHANES-linked mortality data were ascertained through probabilistic record matching with the National Death Index (NDI). Follow-up time per person was calculated beginning at the time of the NHANES interview date and ending at death or censoring. The causes of death were determined from the 10th revision of the International Statistical Classification of Diseases and Related Health Problems [34]. The number of deaths, crude mortality rate and age-standardized mortality rate (ASMRs) of all-cause, cancer and CVD according to quartile of urinary Cd were estimated. ASMRs were based on US Census 2000 population data.

### 2.5. Statistical Analysis

All statistical analyses were performed using SAS software, version 9.4 (SAS Institute Inc., Cary, NC, USA). To account for the complex survey design, the appropriate SAS procedures, weight, strata, and cluster variables were used. Participants were grouped into quartiles based on urinary Cd concentration after adjustment for urinary creatinine and into tertiles based on dietary Zn intake for 1988–1994 and 1999–2004. The frequencies and means of sociodemographic and lifestyle factors according to quartiles of urinary Cd concentration were estimated. Significance testing for the differences in urinary Cd between subgroups was conducted by ANOVA or chi-square test.

Poverty income ratio (PIR) was calculated as a ratio of household income to the poverty threshold after accounting for inflation and family size, based on poverty guidelines provided by the Department of Health and Human Services (HHS). Participants were subsequently dichotomized as PIR ≤1.3 or >1.3. Positive smoking status was defined as having at least 100 lifetime cigarettes. Current smokers were defined as those who did not quit smoking and were stratified as heavy (≥15 cigarettes daily) and light (<15 cigarettes daily). Former smokers were identified as those who had reported quitting smoking by the time of interview. Diabetes mellitus was defined as having a fasting plasma glucose level >126 mg/dl or taking insulin or diabetic medications to lower blood sugar. Hypertension was defined as having a systolic blood pressure >140 mmHg, diastolic pressure >90 mmHg, or taking any prescribed medication for the treatment of high blood pressure. Aspirin users were defined as those using aspirin regularly during the past month. Supplement users were defined as those using nutritional supplements during the past month by the time of the baseline interview.

The interaction between Zn intake and urinary Cd was tested for significance in Cox proportional hazards model. Hazard ratios (HRs), multivariable HRs and 95% confidence intervals (CIs) for all-cause, cancer and CVD mortality were estimated by quartile of urinary Cd, quartile of urinary Cd stratified by level of dietary Zn, and quartile of Cd/Zn ratio. Multivariable model 1 was adjusted for age and gender. Model 2 additionally included ethnicity, body mass index (BMI), PIR, smoking status, diabetes, hypertension, alcohol consumption, saturated fatty acid intake, Zn intake, aspirin use, supplement use and history of CVD and cancer. Alcohol consumption was included as a covariate based on previous studies [35,36]. Covariates were selected for all cause-specific mortalities based on the existing literature. Model 3 of cancer mortality included all variables in Model 2 except hypertension, saturated fatty acid intake and history of CVD [37,38]. Multivariable model 4 of CVD mortality included all variables of Model 2 except Zn intake and history of cancer [39,40,41]. All *p*-values reported are two-sided (α = 0.05).

## 3. Results

When separated into quartiles of Cd exposure using urinary Cd, US adults in the highest quartile were more likely to be women, older, current heavy smokers, and aspirin users compared to those with lower Cd exposures. Those in the highest quartile of Cd exposure also had lower incomes, lower BMIs, and had lower rates of hypertension and histories of cancer, coronary heart disease (CHD) or stroke. Adults in the lowest quartile of urinary Cd were more likely to be men, of younger age, non-smokers, have higher levels of income, and higher intakes of Zn and saturated fat (Table 1).

Of the 5367 total deaths that occurred over a mean follow-up of 15 years, 1194 were attributed to cancer and 1677 were attributed to CVD. Table 2 shows the number of deaths and ASMRs of all-cause, cancer and CVD by quartile of urinary Cd concentration among US adults aged 30 years and older in NHANES 1988–1994 and 1999–2004. A greater urinary Cd concentration was positively associated with ASMRs of all-cause, cancer and CVD.

Table 3 shows Cox proportional HRs and 95% CIs for all-cause, cancer and CVD mortality by quartiles of urinary Cd concentration. In Model 1, after adjusting for age and gender, a positive linear association between urinary Cd and all-cause mortality was observed (HR for the highest versus the lowest quartile 2.04; 95% CI = 1.75 to 2.37; *p*-trend < 0.0001). After further adjustment for potential confounders (Model 2), the association remained significant (HR for the highest versus the lowest quartile 1.38; 95% CI = 1.14 to 1.68; *p*-trend < 0.0001). In multivariable model 1, adjusting for age and gender, we also observed a positive association between urinary Cd and cancer mortality (HR for the highest versus the lowest quartile 2.90; 95% CI = 2.14 to 3.93; *p*-trend < 0.0001) and CVD mortality (HR for the highest versus the lowest quartile 1.68; 95% CI = 1.33 to 2.13; *p*-trend < 0.0001). In multivariable model 3 adjusting for cancer-related potential confounders, we observed a positive linear association between urinary Cd and cancer mortality (HR for the highest versus the lowest quartile 1.54; 95% CI = 1.05 to 2.27; *p*-trend < 0.005). However, after further adjustment for CVD-related potential confounders (Model 4), the association between urinary Cd and CVD mortality was attenuated and no longer significant (HR for the highest versus the lowest quartile 1.22; 95% CI = 0.95 to 1.57; p-trend 0.0502). CVD: Cardiovascular disease.

There was a marginally significant interaction between Zn intake and urinary Cd (*p*-value = 0.08), so we analyzed associations within Zn intake subgroups. Among the lowest tertile of Zn consumers, there was a clear positive association between urinary Cd and cancer mortality (HR for Q4 vs. Q1: 1.79; 95% CI: 1.07–3.01); however, the association was weaker among the highest Zn consumers (HR for Q4 vs. Q1: 1.66; 95% CI: 0.80–3.41) (Table 4). There was no association between urinary Cd and CVD mortality among the lowest tertile and highest tertile of Zn consumers.

Table 5 shows HRs and 95% CIs for all-cause, cancer and CVD mortality by quartile of the Cd/Zn ratio. After adjusting for age and gender, there were positive associations between the Cd/Zn ratio and all-cause mortality (HR for highest vs. lowest quartile (Q4 vs. Q1): 2.17; 95% CI: 1.86–2.52; *p*-trend < 0.0001), cancer mortality (HR for Q4 vs. Q1: 2.86; 95% CI: 2.09–3.92; *p*-trend < 0.0001), and CVD mortality (HR for Q4 vs. Q1: 2.02; 95% CI: 1.60–2.55; *p*-trend < 0.0001). After additional adjustment for related potential confounders, the significance of these positive associations between the Cd/Zn ratio and all-cause mortality (HR for Q4 vs. Q1: 1.54; 95% CI: 1.23–1.93; *p*-trend < 0.0005), cancer mortality (HR for Q4 vs. Q1: 1.65; 95% CI: 1.11–2.47, *p*-trend < 0.05) and CVD mortality (HR for Q4 vs. Q1: 1.49; 95% CI: 1.18–1.88, *p*-trend < 0.001) remained significant.

## 4. Discussion

Using nationally representative data, US adults with the highest urinary Cd tended to be older, female, current heavy smokers, normal weight status and lower levels of income, which is consistent with previous findings based on NHANES 1999–2012 data [42]. Moreover, our findings are in agreement with studies showing that urinary Cd was negatively associated with BMI in NHANES 1999–2002 [15,43]. The current study also found that higher urinary Cd was associated with CVD risk factors such as hypertension, aspirin use, and history of CHD or stroke, consistently with a previous report linking urinary Cd to CVD [44]. We also found that greater urinary Cd was associated with cancer history, which is consistent with previous studies showing that higher urinary Cd level was associated with total cancer, lung cancer [45] and breast cancer risk in Japanese women [46].

This study found that greater urinary Cd is associated with greater all-cause mortality and cancer mortality after adjusting for important potential confounders. These results are consistent with previous studies reporting that urinary Cd levels were associated with a greater risk of all-cause cancer mortality [12,15,47]. Garcia-Esquinas [12] et al. showed that Cd exposure was positively associated with total cancer mortality and with mortality from cancers of the lung and pancreas. While Menke [15] and Larsson [47] reported that urinary Cd was positively associated with CVD mortality, the current study shows that the association between urinary Cd and CVD mortality was diminished after adjusting for all relevant covariates and was marginally significant (HR for the highest versus the lowest quartile 1.22; 95% CI = 0.95 to 1.57; *p*-trend 0.0502).

Several studies reported that Zn can reduce the risk of Cd toxicity by competitive interactions between Zn and Cd [9,16]. Our group previously reported that Zn intake is associated with lower Cd burden [21] and with lower urinary Cd among non-smokers [48]. In our fully adjusted analysis, there was a marginally significant interaction between urinary Cd and Zn intake (*p*-value = 0.08). This finding suggests that the relationship between Zn intake and urinary Cd may differ by Zn intake level, and that both Zn intake and Cd exposure may need to be considered together to precisely predict mortality risk. We therefore investigated the associations of urinary Cd with mortality by Zn intake level. We found that greater urinary Cd was associated with greater cancer mortality only among the lowest tertile of Zn consumers. Because higher Zn intakes appeared to reduce the strength of the association between Cd and mortality, we investigated the combined relationship of Cd exposure and Zn intake with mortality using the Cd/Zn ratio. As the association of urinary Cd with mortality might be modified by dietary Zn level, the Cd/Zn ratio may be a helpful metric for investigating the combined relationship of Cd exposure, Zn intake and mortality. We found a clear positive association between the Cd/Zn ratio cancer mortality, consistent with a previous study by Lin et al. showing a positive association between Cd exposure and cancer mortality among older adults in NHANES III with low Zn intake levels [23]. Lin et al. [23] also showed that the Zn-to-Cd ratio was more strongly associated with cancer mortality risk than either Zn intake or urinary Cd alone. In addition to confirming the association between the Cd/Zn ratio and cancer mortality, the current study also demonstrated that the Cd/Zn ratio is an important predictor of all-cause and CVD mortality. Furthermore, we characterized these associations, showing that linear trends existed between the Cd/Zn ratio and all-cause, cancer and CVD mortality in adjusted models.

This study is strengthened by its use of a large, nationally representative sample of the US adult population. This large and diverse sample allows for the examination of the relationships between urinary Cd, mortality, and Zn intake across participants with varying lifestyle and sociodemographic characteristics. Additionally, this study included a large number of deaths and a long-term follow-up of 15 years. However, this study has several limitations. Firstly, estimation of Zn intake was based on one day of dietary recall data, which may not accurately reflect usual intake of individual participants. Secondly, competing risks can be present between CVD and cancer mortality. Thirdly, even though we tried to match variables between datasets of 1988–1994 and those of 1999–2004, there might be systematic error in combining two datasets. Finally, residual confounding factors might be present in this analysis, although we attempted to adjust for all relevant covariates available in the NHANES dataset.

## 5. Conclusions

In conclusion, a greater Cd/Zn ratio was associated with greater mortality from all causes, cancer, and CVD in fully adjusted models, indicating that the Cd/Zn ratio, rather than urinary Cd or Zn intake alone, may be an important predictor of mortality from all causes, cancer, and CVD. Future research should be conducted to establish recommendation for optimal Zn intake for protecting against Cd toxicity and to explain the relationship between Cd-related disease risk and mortality using biomarkers in a prospective cohort study.

## Figures and Tables

**Table 1 nutrients-12-00056-t001:** Baseline characteristics of US adults ≥ 30 y in NHANES 1988–1994 and 1999–2004 by quartile of urinary Cd concentration (*n* = 15,641).

	NHANES 1988–1994				*p*-Value **	NHANES 1999–2004				*p*-Value **
	Urinary Cd Concentration (µg/g Creatinine) (Range)					Urinary Cd Concentration (µg/g Creatinine) (Range)				
	Q1 *	Q2 *	Q3 *	Q4 *		Q1 *	Q2 *	Q3 *	Q4 *	
	(*n* = 3033)	(*n* = 3034)	(*n* = 3035)	(*n* = 3033)		(*n* = 876)	(*n* = 877)	(*n* = 877)	(*n* = 876)	
	(0.00–0.29)	(0.29–0.52)	(0.52–0.90)	(0.90–23.35)		(0–0.22)	(0.22–0.37)	(0.37–0.62)	(0.62–4.20)	
Follow-up time, mean, years	19.0 (0.2)	17.9 (0.2)	16.9 (0.3)	15.4 (0.2)	<0.0001	9.6 (0.1)	9.5 (0.2)	9.1 (0.1)	8.8 (0.2)	<0.0001
Age, mean, years	43.0 (0.4)	49.1 (0.5)	53.9 (0.6)	57.8 (0.6)	<0.0001	44.7 (0.6)	51.3 (0.6)	54.8 (0.7)	57.8 (0.7)	<0.0001
Men, %	57.2	49.8	43.5	35.8	<0.0001	63.0	45.2	40.0	33.7	<0.0001
Ethnicity					<0.0001					0.073
White, %	86.1	82.3	83.3	86.7		81.5	83.0	84.9	83.1	
Black, %	9.0	13.0	12.3	9.9		11.1	10.6	10.0	11.9	
Mexican-American, %	4.9	4.7	4.4	3.4		7.4	6.4	5.1	4.9	
BMI					<0.0005					<0.005
BMI < 25, %	40.9	37.5	38.1	44.2		27.4	24.1	30.1	39.8	
25 ≤ BMI < 30, %	35.0	35.0	34.4	35.0		39.1	39.9	38.2	34.0	
BMI ≥ 30, %	24.1	27.5	27.5	20.8		33.5	36.0	31.7	26.2	
Smoking ^1^					<0.0001					<0.0001
Never	63.2	48.3	34.5	22.5		67.5	55.8	40.1	24.5	
Former	25.4	32.2	32.9	32.7		24.7	30.0	35.4	34.2	
Current (<15 cigarettes/d)	6.1	7.5	8.4	9.3		5.2	7.6	9.3	10.8	
Current (≥15 cigarettes/d)	5.3	12.0	24.2	34.5		2.6	6.6	15.2	30.5	
PIR ^2^ > 1.3, %	88.7	84.7	82.1	77.7	<0.0001	86.7	83.2	79.2	74.4	<0.0005
Diabetes mellitus ^3^, %	5.7	7.7	8.3	10.4	<0.0001	14.9	14.1	20.0	15.7	0.374
Hypertension ^4^, %	22.2	33.7	40.6	47.6	<0.0001	26.2	37.5	44.0	45.6	<0.0001
Aspirin use ^5^, %	8.4	12.6	15.3	18.4	<0.0001	10.5	13.8	15.6	21.6	<0.001
Supplement use ^6^, %	46.6	43.1	42.2	41.1	0.0695	54.3	59.2	58.8	57.9	0.307
History of CHD^7^ or stroke, %	1.7	3.5	5.8	8.2	<0.0001	2.3	4.9	7.3	10.7	<0.0001
History of cancer, %	4.2	9.2	12.8	13.8	<0.0001	7.6	8.3	11.6	14.4	<0.001
Dietary intake, mean										
Zinc, mg/d	12.7 (0.2)	11.8 (0.2)	11.6 (0.4)	10.7 (0.4)	<0.01	13.5 (0.5)	11.7 (0.3)	11.1 (0.4)	10.7 (0.5)	<0.0005
Alcohol, g/d	10.4 (1.0)	8.7 (0.8)	10.0 (1.2)	9.5 (0.9)	0.112	12.6 (1.4)	10.8 (1.7)	10.1 (1.8)	10.6 (1.7)	0.311
Saturated fats, g/d	29.5 (0.5)	27.0 (0.5)	25.6 (0.8)	24.5 (0.4)	<0.0001	30.3 (1.0)	26.1 (0.8)	24.2 (0.7)	23.3 (0.6)	<0.0001

* First quartile (Q1), second quartile (Q2), third quartile (Q3) and fourth quartile (Q4) ** Tested by ANOVA or chi-square test. ^1^ Smokers defined as those who smoked at least 100 cigarettes in lifetime, former smokers defined as smokers who quit. ^2^ Poverty income ratio. ^3^ Diabetes defined as fasting plasma glucose level greater than 126 mg/dl or taking medication including insulin and/or medication to lower blood sugar. ^4^ Hypertension was defined as systolic blood pressure exceeding 140 mmHg or diastolic pressure over 90 mmHg or taking prescribed medicine for high blood pressure. ^5^ Aspirin users defined as those using aspirin regularly during the past month. ^6^ Defined as those using nutritional supplements during the past month. ^7^ Coronary heart disease (CHD).

**Table 2 nutrients-12-00056-t002:** Age-standardized mortality rates (ASMRs) of all-cause, cancer and CVD among US adults ≥ 30 years old in NHANES 1988–2004 by quartile of urinary Cd concentration ^1^ (*n* = 15,641).

Urinary Cd Concentration	No. of Death	Crude Mortality Rate	Age-Standardized Mortality Rate ^1^
		(per 100,000)	(per 100,000)
All-cause death			
All	5367	1636.5	2357.7
Quartile 1 (lowest)	705	653.8	1918.7
Quartile 2	1062	1202.1	1915.5
Quartile 3	1534	2023.9	2332.8
Quartile 4 (highest)	2066	3065.5	2935.0
Cancer death			
All	1194	389.2	486.9
Quartile 1	139	161.0	343.4
Quartile 2	198	225.6	321.6
Quartile 3	320	460.4	475.9
Quartile 4	537	842.8	733.7
CVD death			
All	1677	483.0	754.4
Quartile 1	229	196.2	680.7
Quartile 2	347	396.7	687.3
Quartile 3	501	611.5	741.2
Quartile 4	600	831.7	840.8

^1^ Age standardization is based on Census 2000 US population data.

**Table 3 nutrients-12-00056-t003:** Multivariable adjusted HR (95% CI) of all-cause, cancer and CVD mortality using Cox proportional hazards models among US adults ≥ 30 years old in NHANES 1988–2004 by quartile of urinary Cd concentration (*n* = 15,641).

	Urinary Cd Concentration (µg/g Creatinine) (Range)				*p* for Trend
	Q1 (*n* = 3910)	Q2 (*n* = 3910)	Q3 (*n* = 3911)	Q4 (*n* = 3910)	
	(0–0.27)	(0.27–0.48)	(0.48–0.82)	(0.82–23.35)	
All-cause					
Model 1 ^1^	1.00	1.13 (0.96–1.33)	1.44 (1.26–1.64)	2.04 (1.75–2.37)	<0.0001
Model 2 ^2^	1.00	0.99 (0.82–1.18)	1.13 (0.96–1.33)	1.38 (1.14–1.68)	<0.0001
Cancer					
Model 1	1.00	1.01 (0.75–1.36)	1.68 (1.31–2.14)	2.90 (2.14–3.93)	<0.0001
Model 3 ^3^	1.00	0.84 (0.60–1.18)	1.21 (0.90–1.62)	1.54 (1.05–2.27)	<0.005
CVD					
Model 1	1.00	1.12 (0.87–1.45)	1.29 (0.99–1.66)	1.68 (1.32–2.13)	<0.0001
Model 4 ^4^	1.00	0.95 (0.73–1.25)	1.01 (0.77–1.33)	1.22 (0.95–1.57)	0.052

^1^ Model 1: Adjusted for age and gender. ^2^ Model 2: Adjusted for age, gender, ethnicity, BMI, PIR, smoking status, diabetes, hypertension, alcohol consumption, saturated fatty acid intake, Zn intake, aspirin use, supplement use and history of CVD and cancer. ^3^ Model 3: Adjusted for age, gender, ethnicity, BMI, PIR, smoking status, diabetes, alcohol consumption, Zn intake, aspirin use, supplement use and history of cancer. ^4^ Model 4: Adjusted for age, gender, ethnicity, BMI, PIR, smoking status, diabetes, hypertension, alcohol consumption, saturated fatty acid intake, aspirin use, supplement use and history of CVD.

**Table 4 nutrients-12-00056-t004:** Multivariable adjusted HR (95% CI) of all-cause, cancer and CVD mortality using Cox proportional hazards models among US adults ≥ 30 years old in NHANES 1988–2004 by quartile of urinary Cd concentration by tertile of Zn intake (*n* = 15,641).

**Urinary Cd Concentration (µg/g Creatinine) (Range)**					
	**1st Tertile of Dietary Zn Intake**				***p*** **for trend**
	**Q1 (*n* = 1303)**	**Q2 (*n* = 1304)**	**Q3 (*n* = 1304)**	**Q4 (*n* = 1303)**	
	**(0.00–0.31)**	**(0.31–0.54)**	**(0.54–0.93)**	**(0.93–16.17)**	
All-cause					
Model 1 ^1^	1.00	1.03 (0.82–1.30)	1.29 (1.04–1.58)	1.73 (1.39–2.15)	<0.0001
Model 2 ^2^	1.00	1.00 (0.80–1.25)	1.04 (0.81–1.34)	1.25 (0.96–1.63)	0.062
Cancer					
Model 1	1.00	1.10 (0.63–1.91)	2.01 (1.13–3.54)	3.71 (2.24–6.14)	<0.0001
Model 3 ^3^	1.00	0.86 (0.49–1.52)	1.25 (0.66–2.36)	1.79 (1.07–3.01)	<0.005
CVD					
Model 1	1.00	1.01 (0.73–1.39)	1.11 (0.81–1.52)	1.17 (0.83–1.65)	0.293
Model 4 ^4^	1.00	0.92 (0.61–1.39)	0.97 (0.65–1.47)	0.96 (0.59–1.55)	0.958
	**2nd tertile of dietary Zn intake**				
	**Q1 (*n* = 1293)**	**Q2 (*n* = 1292)**	**Q3 (*n* = 1294)**	**Q4 (*n* = 1293)**	
	**(0.00–0.27)**	**(0.27–0.47)**	**(0.47–0.81)**	**(0.81–21.65)**	
All-cause					
Model 1 ^1^	1.00	1.15 (0.91–1.46)	1.33 (1.08–1.64)	1.94 (1.55–2.44)	<0.0001
Model 2 ^2^	1.00	0.98 (0.75–1.30)	0.95 (0.75–1.20)	1.25 (0.94–1.66)	0.067
Cancer					
Model 1	1.00	1.79 (1.07–2.98)	2.41 (1.59–3.68)	3.80 (2.34–6.17)	<0.0001
Model 3 ^3^	1.00	1.47 (0.83–2.60)	1.45 (0.96–2.21)	1.79 (0.95–3.37)	0.070
CVD					
Model 1	1.00	1.10 (0.71–1.69)	1.24 (0.79–1.96)	1.73 (1.18–2.54)	<0.001
Model 4 ^4^	1.00	1.18 (0.70–1.98)	1.17 (0.70–1.94)	1.49 (0.89–2.49)	0.102
	**3rd tertile of dietary Zn intake**				
	**Q1 (*n* = 1312)**	**Q2 (*n* = 1315)**	**Q3 (*n* = 1314)**	**Q4 (*n* = 1314)**	
	**(0–0.23)**	**(0.23–0.42)**	**(0.42–0.74)**	**(0.74–23.35)**	
All-cause					
Model 1 ^1^	1.00	1.09 (0.83–1.43)	1.62 (1.22–2.14)	2.38 (1.89–3.04)	<0.0001
Model 2 ^2^	1.00	0.94 (0.68–1.32)	1.43 (1.01–2.04)	1.71 (1.23–2.38)	<0.0001
Cancer					
Model 1	1.00	0.71 (0.38–1.33)	1.29 (0.75–2.23)	2.22 (1.30–3.77)	<0.0005
Model 3 ^3^	1.00	0.65 (0.29–1.42)	1.23 (0.61–2.47)	1.66 (0.80–3.41)	<0.05
CVD					
Model 1	1.00	1.10 (0.62–1.94)	1.48 (0.81–2.69)	2.30 (1.31–4.02)	<0.0005
Model 4 ^4^	1.00	0.75 (0.41–1.35)	0.93 (0.51–1.67)	1.32 (0.77–2.26)	0.081

^1^ Model 1: Adjusted for age and gender. ^2^ Model 2: Adjusted for age, gender, ethnicity, BMI, PIR, smoking status, diabetes, hypertension, alcohol consumption, saturated fatty acid intake, Zn intake, aspirin use, supplement use and history of CVD and cancer. ^3^ Model 3: Adjusted for age, gender, ethnicity, BMI, PIR, smoking status, diabetes, alcohol consumption, Zn intake, aspirin use, supplement use and history of cancer. ^4^ Model 4: Adjusted for age, gender, ethnicity, BMI, PIR, smoking status, diabetes, hypertension, alcohol consumption, saturated fatty acid intake, aspirin use, supplement use and history of CVD.

**Table 5 nutrients-12-00056-t005:** Multivariable adjusted HR (95% CI) of all-cause, cancer and CVD mortality using Cox proportional hazards models among US adults ≥ 30 years old in NHANES 1988–2004 by quartile of Cd/Zn ratio (*n* = 15,641).

	1988–2004				*p* for Trend
	Cd/Zn Ratio (Range)				
	Q1 (*n* = 3910)	Q2 (*n* = 3910)	Q3 (*n* = 3911)	Q4 (*n* = 3910)	
	(0–0.002)	(0.002–0.005)	(0.005–0.011)	(0.011–6.464)	
All-cause					
Model 1 ^1^	1.00	1.36 (1.16–1.60)	1.68 (1.43–1.96)	2.17 (1.86–2.52)	<0.0001
Model 2 ^2^	1.00	1.27 (1.04–1.57)	1.29 (1.05–1.59)	1.54 (1.23–1.93)	<0.0005
Cancer					
Model 1	1.00	1.43 (1.06–1.92)	1.87 (1.36–2.59)	2.86 (2.09–3.92)	<0.0001
Model 3 ^3^	1.00	1.27 (0.91–1.76)	1.33 (0.90–1.96)	1.65 (1.11–2.47)	<0.05
CVD					
Model 1	1.00	1.24 (0.95–1.62)	1.71 (1.30–2.26)	2.02 (1.60–2.55)	<0.0001
Model 4 ^4^	1.00	1.17 (0.86–1.58)	1.28 (0.99–1.67)	1.49 (1.18–1.88)	<0.001

^1^ Model 1: Adjusted for age and gender. ^2^ Model 2: Adjusted for age, gender, ethnicity, BMI, PIR, smoking status, diabetes, hypertension, alcohol consumption, saturated fatty acid intake, aspirin use, supplement use and history of CVD and cancer. ^3^ Model 3: Adjusted for age, gender, ethnicity, BMI, PIR, smoking status, diabetes, alcohol consumption, aspirin use, supplement use and history of cancer. ^4^ Model 4: Adjusted for age, gender, ethnicity, BMI, PIR, smoking status, diabetes, hypertension, alcohol consumption, saturated fatty acid intake, aspirin use, supplement use and history of CVD.

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
