# Peer review of "Association between Urinary Cadmium-to-Zinc Intake Ratio and Adult Mortality in a Follow-Up Study of NHANES 1988–1994 and 1999–2004"

_nutrients, 2019, doi:10.3390/nu12010056_

Round 1
Reviewer 1 Report
The paper describes a cross-sectional study (combined NHANES 1988–1994 and 1999–2004) that followed for a mean of 15 years for comparing the mortality rates of all-cause, cancer and CVD according to quartile of urinary Cd, and finds that urinary Cd was positively associated with all-cause mortality and cancer mortality. However, the association might be altered by the levels of dietary Zn intake. The authors use urinary Cd to dietary Zn ratio as a predictor and show the positive association with mortality from all causes, cancer, and CVD.
Major comments: This study confirms previous results on this topic but using different predictor (this study: the ratio of urinary Cd to dietary Zn vs. previous studies: the ratio of Zn to Cd). Why do authors prefer to use this indicator? Reasons should be stated in the manuscript.
Minor comment 1: How did you choose variables to adjust in the Cox proportional hazards models? Why did the models include variable of alcohol consumption?
Minor comment 2: Results: the paragraph 1 of Characteristics of the survey population section need to be clarified and re-written.
Minor comment 3: Discussion: The authors discuss the conclusions in light of other research in detail and cite these references. These findings are consistent with previous work based on data from NHANES study or other previous studies. Based on the results of the study, what new findings would authors add?
Author Response
Response to the First Reviewer’s Comments
The paper describes a cross-sectional study (combined NHANES 1988–1994 and 1999–2004) that followed for a mean of 15 years for comparing the mortality rates of all-cause, cancer and CVD according to quartile of urinary Cd, and finds that urinary Cd was positively associated with all-cause mortality and cancer mortality. However, the association might be altered by the levels of dietary Zn intake. The authors use urinary Cd to dietary Zn ratio as a predictor and show the positive association with mortality from all causes, cancer, and CVD.
1. Major comments: This study confirms previous results on this topic but using different predictor (this study: the ratio of urinary Cd to dietary Zn vs. previous studies: the ratio of Zn to Cd). Why do authors prefer to use this indicator? Reasons should be stated in the manuscript.
A preliminary study by our research group showed that dietary and serum Zn in US adults are associated with Cd exposure1. We also found that there was a marginally significant interaction between urinary Cd and Zn intake (P-value=0.08). We aim to explore how to reduce the risk of Cd exposure by modifying Zn intake. As the association of urinary Cd with mortality might be modified by Zn intake level, we think that it is reasonable to use the Cd/Zn ratio to investigate the combined relationship of Cd exposure and Zn intake with mortality. We added more explanation about this concept in the discussion section. (Page 9, lines 236-256) We have seen in the existing literature that variations of both a Cd/Zn ratio and Zn/Cd ratio have been used for differing research purposes over the last several years. This is still an emerging area of research, and this ratio seems to be more of an exploratory metric rather than a standard measurement. Therefore, we chose to evaluate Cd/Zn ratio based on the theoretical framework that Cd (numerator) was a key predictor of mortality and that it may be modified by Zn levels (denominator).
2. Minor comment 1: How did you choose variables to adjust in the Cox proportional hazards models? Why did the models include variable of alcohol consumption?
Table1 shows that urinary Cd is associated with variables including age, gender, ethnicity, BMI, smoking status, income level, diabetes, aspirin use, history of CHD or stroke, history of cancer, and intakes of zinc and saturated fat. Although we did not see a significant association between urinary Cd and alcohol consumption in our analysis, we included this variable because several studies have reported that alcohol consumption might be an important variable for the association with mortality2,3.
3. Minor comment 2: Results: the paragraph 1 of Characteristics of the survey population section need to be clarified and re-written.
We have revised this paragraph to present the same key information with greater clarity. We feel it is now easier to read and understand (Page 3, lines 132-137).
4. Minor comment 3: Discussion: The authors discuss the conclusions in light of other research in detail and cite these references. These findings are consistent with previous work based on data from NHANES study or other previous studies. Based on the results of the study, what new findings would authors add?
Compared to the previous study, this study showed that the ratio of urinary Cd to Zn could be a good indicator for all-cause mortality and CVD mortality as well as cancer mortality. Importantly, we also showed that dietary Zn intake modified the association between Cd exposure and mortality from all-cause, cancer, and CVD.
References
1. Vance TM, Chun OK. Zinc Intake Is Associated with Lower Cadmium Burden in U.S. Adults. The Journal of nutrition. 2015;145(12):2741-2748.
2. Ronksley PE, Brien SE, Turner BJ, Mukamal KJ, Ghali WA. Association of alcohol consumption with selected cardiovascular disease outcomes: a systematic review and meta-analysis. BMJ. 2011;342:d671.
3. Xi B, Veeranki SP, Zhao M, Ma C, Yan Y, Mi J. Relationship of Alcohol Consumption to All-Cause, Cardiovascular, and Cancer-Related Mortality in U.S. Adults. J Am Coll Cardiol. 2017;70(8):913-922.
Reviewer 2 Report
Title and summary (clarity and structure)
The title should be clearer and more concise, for example, "Association between urinary cadmium / zinc intake ratio with adult mortality in NHANES 1988-1994 and 1999-2004". The conclusion in summary does not match the results of this study. In keywords it would be better to change: zinc intake; ratio urinary Cd / intake Zn.
Relevance of the topic, originality of work and review of the literature.
This study is important and interesting. However, the introduction, the material and the methods are not sufficient in its description. It would be useful if the authors can make an adequate description of each point studied. The explanation of the importance of this study should be deepened. The comparison with other studies is also regular. There are other studies that find a similar association and the authors do not indicate the importance of these results.
Structure and organization of the article. Argumentative ability. Drafting
The structure of this research is appropriate. However, the introduction is poor. It should be improved. The manuscript is written correctly, is clear and easy to read, but must be improved. Although they have 40 citations in their references, it is too concise and lacks basic information to understand the importance of the results. In addition, it is extremely little argumentative. The results are well characterized in tables.
Methodological rigor Research instruments
The methods are replicable. However, the authors must deepen the methodology. Why did the authors not use people under 30 and what is the age limit of the participants? Why didn't they consider gender important? What is the toxic value of cadmium in the blood, in the urine for humans? Are the mentioned values in the toxicity range? What is the value of zinc consumption in each tertile? The best known biomarker for cadmium toxicity is urinary cadmium. Why do the authors use mg / d instead of urinary cadmium / gr creatinine? The reason why the authors used different ranges of reference should be described in the discussion.
Research results. Advances. Discussion. Conclusions.
If the main objective was to describe the mortality rate that occurred during the follow-up of the participants during NHANES 1988-2004, in relation to the urinary cadmium/zinc intake ratio, it would be important to describe the percentage of patients with mortality due to cancer, cardiovascular and those who only presented the disease. The results of zinc intake and mortality did not describe in this manuscript.
In the conclusion…indicating that the ratio of urinary Cd to dietary Zn rather than urinary Cd or Zn intake alone may be an important predictor of mortality from all causes, cancer, and CVD. The comparison between zinc consumption (alone) associated with mortality is not clearly observed in the manuscript. The authors must bear in mind that they try to describe two different moments, and the differences in the methodology must be described, and they must justify why they did not use the data before and after the years studied.
The results seem plausible and there is enough data.
The analysis is correct but the discussion should be improved. The authors must explain the Zn-a-Cd relationship used in NHANES III, why they are different or not from the study carried out by them.
The conclusions are not consistent with the evidence and arguments presented. The conclusions should reflect the objectives achieved or not.
Trends should support discussion and conclusions.
Suggestions should be written at the end of the discussion.
The references are precise, adequate and balanced, but the authors should take better advantage of them.
The following changes should be considered:
In keywords, it is better: urinary Cd/Zn intake ratio.
Improve the description of the material and methods, the results, deepen the analysis and be more descriptive and argumentative.
The results are interesting. A better discussion of the significant results should be made.
The conclusions should be inferred only from the results obtained.

Author Response
The authors appreciate the thorough and constructive comments of the reviewers, which have helped us to greatly improve this paper. The accompanying manuscript has been reformulated incorporating suggestions from the Nutrientsreviewers.
Response to the Second Reviewer’s Comments
Title and summary (clarity and structure)
1. The title should be clearer and more concise, for example, "Association between urinary cadmium / zinc intake ratio with adult mortality in NHANES 1988-1994 and 1999-2004". The conclusion in summary does not match the results of this study. In keywords it would be better to change: zinc intake; ratio urinary Cd / intake Zn.
As advised, we revised the title to “Association between urinary cadmium to zinc intake ratio with adult mortality in a follow-up study of NHANES 1988-1994 and 1999-2004”. We also changed the keywords to ‘zinc intake, urinary Cd/Zn intake ratio’. As advised, we revised the conclusion in the abstract. Thank you. (Page 1, lines 2-3, 26-28)
Relevance of the topic, originality of work and review of the literature.
2. This study is important and interesting. However, the introduction, the material and the methods are not sufficient in its description. It would be useful if the authors can make an adequate description of each point studied. The explanation of the importance of this study should be deepened. The comparison with other studies is also regular. There are other studies that find a similar association and the authors do not indicate the importance of these results.
Thank you for your constructive comment. We have revised the introduction to further explain the rationale for this study and to describe the mechanistic basis for our hypothesis. We also extensively revised the methods section to make sure all variables were clearly defined and their use was justified. We also edited the text of the results to make sure that our findings were clearly stated and easy to read. Finally, the discussion has been revised to more explicitly state the importance of our work and how it expands upon the existing literature, particularly in demonstrating an association between the Cd/Zn ratio and all-cause and CVD mortality.
Structure and organization of the article. Argumentative ability. Drafting
3. The structure of this research is appropriate. However, the introduction is poor. It should be improved. The manuscript is written correctly, is clear and easy to read, but must be improved. Although they have 40 citations in their references, it is too concise and lacks basic information to understand the importance of the results. In addition, it is extremely little argumentative. The results are well characterized in tables.
Thank you for this feedback. As described above, the introduction has been improved to clarify the relationship between Zn and Cd and explain the basis for our research approach. We also revised the discussion to more clearly interpret our results, underscore the similarities with previous work, and explain the novel findings of this research (Page 2, lines 49-52; Page 9, lines 222-224, 238-256).
Methodological rigor Research instruments
4. The methods are replicable. However, the authors must deepen the methodology. Why did the authors not use people under 30 and what is the age limit of the participants? Why didn't they consider gender important? What is the toxic value of cadmium in the blood, in the urine for humans? Are the mentioned values in the toxicity range? What is the value of zinc consumption in each tertile? The best-known biomarker for cadmium toxicity is urinary cadmium. Why do the authors use mg / d instead of urinary cadmium / gr creatinine? The reason why the authors used different ranges of reference should be described in the discussion.
As incidence and deaths from cancer or CVD are very low for young people, it would be reasonable to exclude young people for the mortality study. Based on data in the statistics of cancer and CVD deaths1,2, we excluded subjects under 30. We adjusted gender as a covariate in this analysis.
While elevated blood Cd confirms recent acute exposure, it does not correlate well with body burden.3On the other hand, urinary Cd reflects integrated exposure and total body burden. Microscopic tubular proteinuria is known to occur at urinary Cd levels of 2 μg/g creatinine, and renal dysfunction is likely to occur at urinary Cd levels greater than 10 µg/g creatinine4,5. We added an explanation in the Introduction (Page 2, lines 49-51). We used creatinine adjusted urinary Cd concentration in this analysis and corrected our unit mistake. Thank you.
Research results. Advances. Discussion. Conclusions.
5. If the main objective was to describe the mortality rate that occurred during the follow-up of the participants during NHANES 1988-2004, in relation to the urinary cadmium/zinc intake ratio, it would be important to describe the percentage of patients with mortality due to cancer, cardiovascular and those who only presented the disease.
As the raw percentage of deaths depends on age distribution, we presented age-standardized mortality based on Census 2000 US population data. We reported all-cause death numbers, crude mortality rate and age-standardized mortality in Table 2.
6. The results of zinc intake and mortality did not describe in this manuscript. In the conclusion…indicating that the ratio of urinary Cd to dietary Zn rather than urinary Cd or Zn intake alone may be an important predictor of mortality from all causes, cancer, and CVD. The comparison between zinc consumption (alone) associated with mortality is not clearly observed in the manuscript. The authors must bear in mind that they try to describe two different moments, and the differences in the methodology must be described, and they must justify why they did not use the data before and after the years studied.
A preliminary study by our research group showed that dietary and serum Zn in US adults is associated with Cd exposure6. We also found that there was a marginally significant interaction between urinary Cd and Zn intake (P-value=0.08). This result suggests that the relationship between Zn intake and urinary Cd may differ by Zn intake level. Therefore, Zn intake and Cd exposure may need to be considered together to predict mortality risks. We aimed to know how to reduce the risk of Cd exposure using a dietary modifier of Zn intake. As the association of urinary Cd with mortality might be modified by Zn intake level, we investigated the combined relationship of Cd exposure, Zn intake and mortality using the Cd/Zn ratio. We added the explanation in the discussion section. (Page 9, lines 236-256)
7. The results seem plausible and there is enough data. The analysis is correct but the discussion should be The authors must explain the Zn-a-Cd relationship used in NHANES III, why they are different or not from the study carried out by them.
As mentioned in the discussion (page 9, lines 248-250), our findings were consistent with a previous study using NHANES III data. Lin et at (ref 24) showed that cadmium exposure was associated with an increased risk of cancer mortality among older adults with low zinc intake. Similarly, our results showed that higher Cd in the setting of low Zn intake (high Cd/Zn ratio) was predictive of increased cancer risk. We slightly revised the wording in this section of the discussion to improve clarity.
8. The conclusions are not consistent with the evidence and arguments presented. The conclusions should reflect the objectives achieved or not. Trends should support discussion and conclusions. Suggestions should be written at the end of the discussion. The references are precise, adequate and balanced, but the authors should take better advantage of them.
We revised the conclusion in the abstract. As advised, we added suggestion in the end of discussion (Page 10, lines 270-272).
The following changes should be considered:
9. In keywords, it is better: urinary Cd/Zn intake ratio.
As advised, we changed the keywords to ‘urinary Cd/Zn intake ratio’. Thank you.
10. Improve the description of the material and methods, the results, deepen the analysis and be more descriptive and argumentative. The results are interesting. A better discussion of the significant results should be made. The conclusions should be inferred only from the results obtained.
Thank you for this feedback. As described briefly above, we have made major revisions to the methods, results, and discussion to improve clarity. With your suggestions, we feel that we have now significantly improved our manuscript by explaining our findings and their interpretations in the context of the existing body of literature.
References
U.S. Cancer Statistics Working Group. U.S. Cancer Statistics Data Visualizations Tool, based on November 2018 submission data (1999-2016). In: U.S. Department of Health and Human Services, Centers for Disease Control and Prevention and National Cancer Institute; June 2019. Kenneth D. Kochanek, M.A. SLM, Jiaquan Xu, Elizabeth Arias. National Vital Statistics Reports. Hyattsville, MD: U.S. DEPARTMENT OF HEALTH AND HUMAN SERVICES, Centers for Disease Control and Prevention, National Center for Health Statistics;June 24, 2019. Jarup L. Cadmium overload and toxicity. Nephrol Dial Transplant. 2002;17 Suppl 2:35-39. Agency for Toxic Substances and Disease Registry. TOXICOLOGICAL PROFILE FOR CADMIUM. https://www.atsdr.cdc.gov/ToxProfiles/tp5.pdf. Published September 2012. Accessed October, 2018. Roels HA, Hoet P, Lison D. Usefulness of biomarkers of exposure to inorganic mercury, lead, or cadmium in controlling occupational and environmental risks of nephrotoxicity. Ren Fail. 1999;21(3-4):251-262. Vance TM, Chun OK. Zinc Intake Is Associated with Lower Cadmium Burden in U.S. Adults. The Journal of nutrition. 2015;145(12):2741-2748.Round 2
Reviewer 2 Report
To publish this manuscript, authors should review the suggested changes in the attached pdf.
The order of references is not right. The authors did not write new four references.
The authors should add in material and methods: Since the incidence and deaths from cancer or CVD are very low for young people, those under 30 years of age were excluded from this mortality study. Gender was adjusted as a covariate in this analysis.
What units were used for urinary Cd and Zn intake?
What are the recommended doses for zinc intake and the toxic level for urine Cd concentration?
The authors should add in their discussion:
Although the daily dose of Zn intake not meeting the European recommendations for zinc
Author Response
The authors appreciate the constructive comments of the reviewers, which have helped us to greatly improve this paper. The accompanying manuscript has been revised incorporating suggestions from the Nutrients reviewers. Some parts in the introduction, method, and discussion section were re-phrased for clarifying the meaning.
To publish this manuscript, authors should review the suggested changes in the attached pdf.
1. The order of references is not right. The authors did not write new four references.
We corrected the order of all references and added new references. Thank you!
2. The authors should add in material and methods: Since the incidence and deaths from cancer or CVD are very low for young people, those under 30 years of age were excluded from this mortality study. Gender was adjusted as a covariate in this analysis.
As advised, we added the sentence in methods (Page 2, lines 70-71). We addressed that the covariates including gender were adjusted in Models in the method section. (Page 3, line 119)
3. What units were used for urinary Cd and Zn intake? What are the recommended doses for zinc intake and the toxic level for urine Cd concentration? The authors should add in their discussion: Although the daily dose of Zn intake not meeting the European recommendations for zinc.
The unit of urinary Cd and zinc intake are µg/g creatinine and mg/day. As we stated in the introduction, the toxic level of urine Cd was 2 µg/g creatinine for microscopic tubular proteinuria and 10 µg/g creatinine for renal dysfunction. Although recommended dietary allowances of Zn by Food and Nutrition Board of the Institute of Medicine, National Academy of Sciences are 11 mg/d for men and 8 mg/d for women, we cannot establish the recommended dosage for protecting Cd toxicity from this study. We addressed this in the discussion section (Page 10, lines 265). Thank you!